# PROTEIN DISCOVERY WITH DISCRETE WALK-JUMP SAMPLING

**Nathan C. Frey**[1,*,†]    **Daniel Berenberg**[1,3,*]    **Karina Zadorozhny**[1]    **Joseph Kleinhenz**[1]
**Julien Lafrance-Vanasse**[2]    **Isidro Hötzel**[2]    **Yan Wu**[2]
**Stephen Ra**[1]    **Richard Bonneau**[1]    **Kyunghyun Cho**[1,3,4]
**Andreas Loukas**[1]    **Vladimir Gligorijević**[1]    **Saeed Saremi**[1,†]
[1]Prescient Design, Genentech    [2]Antibody Engineering, Genentech
[3]Department of Computer Science, New York University
[4]Center for Data Science, New York University

## ABSTRACT

We resolve difficulties in training and sampling from a discrete generative model by learning a smoothed energy function, sampling from the smoothed data manifold with Langevin Markov chain Monte Carlo (MCMC), and projecting back to the true data manifold with one-step denoising. Our *Discrete Walk-Jump Sampling* formalism combines the contrastive divergence training of an energy-based model and improved sample quality of a score-based model, while simplifying training and sampling by requiring only a single noise level. We evaluate the robustness of our approach on generative modeling of antibody proteins and introduce the *distributional conformity score* to benchmark protein generative models. By optimizing and sampling from our models for the proposed distributional conformity score, 97-100% of generated samples are successfully expressed and purified and 70% of functional designs show equal or improved binding affinity compared to known functional antibodies on the first attempt in a single round of laboratory experiments. We also report the first demonstration of long-run fast-mixing MCMC chains where diverse antibody protein classes are visited in a single MCMC chain.

## 1 INTRODUCTION

Discrete sequence generation poses a number of challenges to gradient-based generative models. Generative models must be expressive enough to faithfully capture the underlying data distribution, while also having controllable outputs that are novel, unique, diverse, and respect the constraints of the problem space. Energy-based models (EBMs) (Hinton & Sejnowski, 1986; LeCun et al., 2006) fit an energy function that specifies a probability distribution over data analogous to the Boltzmann distribution from statistical physics. Giving access to an easily computable energy is an advantage of EBMs, but on the flip-side they can be difficult to train and sample from. Denoising objectives based on score matching (Hyvärinen, 2005; Vincent, 2011) and the related advancements in diffusion models (Sohl-Dickstein et al., 2015; Ho et al., 2020) overcome these issues, but these either model the energy gradient or only provide access to an empirical lower-bound of the likelihood.

Protein design is an instance of the discrete sequence generation problem, wherein the challenge is to find useful proteins in the large, discrete, and sparsely functional space (Romero & Arnold, 2009) of dimension $20^L$ for proteins of length $L$. Here, we consider the specific problem of generative modeling of antibodies, a class of proteins with highly conserved structure that are of immense interest for therapeutics. In addition to the qualities mentioned above, generative models for antibodies must be sample-efficient because of the relatively small size of datasets with therapeutic antibodies (Kim et al., 2023). Antibodies consist of well-conserved domains and high-entropy variable regions, so leveraging evolutionary information from pre-trained protein language models is not an immediate solution. We distinguish *ab initio* protein discovery and design (producing novel, functional proteins

---

*Equal contribution
†Correspondence to: {frey.nathan.nf1, saremi.saeed}@gene.com

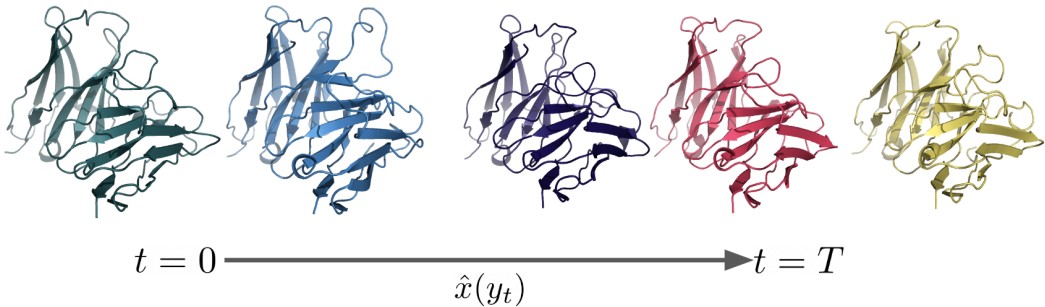

Figure 1: Selected samples from a single Markov chain Monte Carlo sampling run of discrete Walk-Jump sampling (our method). Protein color corresponds to different antibody germlines (classes). Samples are folded with EquiFold (Lee et al., 2022) for visualization purposes. Discrete walk-jump sampling exhibits fast mixing and explores diverse modes of the distribution in a single chain.

given some training samples), which is the focus of this work, from *de novo* design, which we define as the generation of novel proteins without starting material. Existing autoregressive protein design methods (Jin et al., 2021) are inefficient and can suffer from accumulation of errors and high inference latency, while current non-autoregressive diffusion models are similarly inefficient and poorly optimized for real discovery and design tasks (Kong et al., 2022). Our goal here is to invent an efficient, non-autoregressive generative modeling paradigm for discrete data that produces high quality, novel samples.

To this end, we introduce **S**moothed **D**iscrete **S**ampling (**SDS**), a new formalism for training and sampling from discrete generative models. We propose a novel algorithm, **d**iscrete **W**alk-**J**ump **S**ampling (**dWJS**), a method building on the neural empirical Bayes (NEB) (Saremi & Hyvärinen, 2019) formalism, that addresses the brittleness of discrete EBMs and diffusion models and in doing so, provides a robust and general framework for protein discovery and design.[1] We also design a metric called the **D**istributional **C**onformity **S**core (**DCS**), which is a simple scalar score for protein sample quality. Our results rescue EBMs for discrete distribution modeling and question the need for diffusion models with multiple noise scales in protein discovery.

Our contributions are as follows:

- We introduce a new paradigm for modeling discrete data distributions, *Smoothed Discrete Sampling* (SDS), building on the neural empirical Bayes framework. We propose the discrete Walk-Jump sampling algorithm, which uses uncoupled, separately trained score- and energy-based models to learn *noisy* data distributions and sample discrete data. dWJS enables fast, non-autoregressive sampling with variable length discrete outputs. We also design a novel architecture for discrete EBMs.

- Our method simplifies score-based model training for discrete data by requiring only a single noise level and no noise schedule, which alleviates the brittleness, training instabilities, and slow sampling of diffusion models. Our method also resolves difficulties in training EBMs, obviating the need for many common EBM training tricks (replay buffer, $\ell_2$ norm penalty, rejection sampling, etc.), while preserving good sample quality and fast sampling.

- We demonstrate the utility of our approach in the context of *ab initio* protein discovery and design - generating novel, biophysically-valid protein sequences from models trained on repertoires of functional molecules. We validate our method with *in vitro* experiments. Our method outperforms autoregressive and masked protein language models, large language models, discrete sequence and structure-sequence diffusion, and score-based baselines.

---

[1]https://github.com/prescient-design/walk-jump

## 2 BACKGROUND

### 2.1 ENERGY-BASED MODELS

EBMs are a class of models that learn an energy function $f_\theta : \mathcal{X} \to \mathbb{R}$ mapping inputs $x$ (in $\mathbb{R}^d$) to a scalar "energy" value. The data distribution is approximated by the Boltzmann distribution

$$p_\theta(x) \propto e^{-f_\theta(x)}.$$

EBMs are typically trained via contrastive divergence (Hinton, 2002), and new samples are drawn from $p_\theta(x)$ by Markov-Chain Monte Carlo (MCMC). Details of the loss function used in this work are given in Section 3. In Langevin MCMC, samples are initialized from a known data point or random noise and refined with (discretized) Langevin diffusion

$$x_{k+1} = x_k - \delta \nabla f_\theta(x_k) + \sqrt{2\delta}\, \varepsilon_k, \ \varepsilon_k \sim \mathcal{N}(0, I_d), \tag{1}$$

where $\nabla$ denotes the gradient of the energy function with respect to inputs, $k$ is the sampling step, $\delta$ is the (discretization) step size, and the noise $\varepsilon_k$ is drawn from the normal distribution at each step.

### 2.2 NEURAL EMPIRICAL BAYES

In NEB, the random variable $X$ is transformed with additive Gaussian noise $Y = X + \mathcal{N}(0, \sigma^2 I_d)$. The least-squares estimator of $X$ given $Y = y$ is given by (Robbins, 1956; Miyasawa, 1961)

$$\hat{x}(y) = y + \sigma^2 \nabla \log p(y), \tag{2}$$

where $p(y) = \int p(y|x)p(x)dx$ is the probability distribution function of the smoothed density.[2] This estimator is often expressed directly in terms of $g(y) = \nabla \log p(y)$ known as the score function (Hyvärinen, 2005) which is parameterized with a neural network denoted by $g_\phi : \mathbb{R}^d \to \mathbb{R}^d$. The least-squares estimator then takes the following parametric form:

$$\hat{x}_\phi(y) = y + \sigma^2 g_\phi(y). \tag{3}$$

Putting this all together leads to the following learning objective

$$\mathcal{L}(\phi) = \mathbb{E}_{x \sim p(x), y \sim p(y|x)} \| x - \hat{x}_\phi(y) \|^2, \tag{4}$$

which is optimized with stochastic gradient descent. Notably, no MCMC sampling is required during learning. In short, the objective is "learning to denoise" with an empirical Bayes formulation (discussed further in Appendix B.3).

## 3 ANTIBODY DISCOVERY AND DESIGN

### 3.1 DISCRETE WALK-JUMP SAMPLING

Following training of the denoising network, $g_\phi$, one can sample noisy data using the learned score function $g_\phi(y)$ with Langevin MCMC (replace $-\nabla f$ with $g$ in Eq. 1). For any such draws $y_k$, clean samples from the true data manifold $\mathcal{M}$ are obtained by "jumping" back to $\mathcal{M}$ with the least-squares estimator $\hat{x}_\phi(y_k) = y_k + \sigma^2 g_\phi(y_k)$. This is the walk-jump sampling (WJS) scheme. A key property of WJS is the fact that the least-squares estimation (jump) is *decoupled* from the Langevin MCMC (walk).

Here, we take advantage of this decoupling to train an EBM with maximum likelihood estimation on the smoothed distribution of noisy sequences, generate noisy samples with Langevin MCMC, and denoise samples with a separately trained neural network, the least-squares estimator. The algorithm for discrete walk-jump sampling is given in Algo. 1. Our algorithm is general and applies to any discrete sequence inputs of a fixed vocabulary. In Fig. 1 we show samples generated from a single chain of MCMC. Unlike a diffusion model, every sample along the chain collectively forms a valid set of samples from the underlying distribution, because of the decoupled walk (sampling) and jump (denoising) steps. dWJS also produces fast-mixing chains, such that many diverse modes (protein

---

[2]We follow the convention $p(x) \coloneqq p_X(x), p(y) \coloneqq p_Y(y)$, etc.

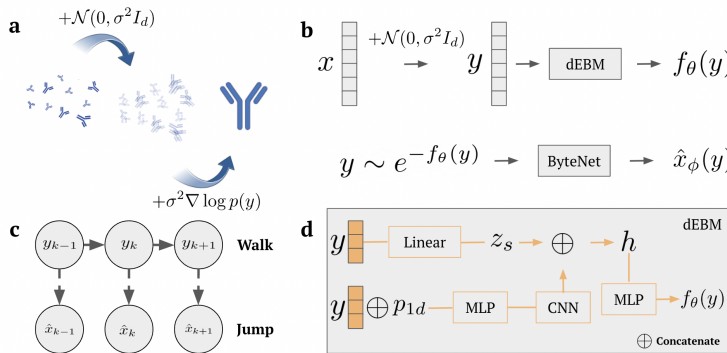

Figure 2: Discrete walk-jump sampling. **a** The noising and denoising process is applied to antibody proteins. **b** Discrete inputs $x$ are smoothed with isotropic Gaussian noise, $\varepsilon \sim \mathcal{N}(0, \sigma^2 I_d)$, to noisy inputs, $y = x + \varepsilon$. A discrete energy-based model (dEBM) parameterizes the energy function $f_\theta(y)$ of noisy data. Noisy data is sampled with the energy function, and denoised with a separate denoising ByteNet network to clean samples, $\hat{x}_\phi(y)$. **c** The "walk" sampling steps on the noisy data manifold with Langevin MCMC are totally decoupled from the "jump" steps to clean samples. **d** The dEBM takes noisy inputs $y$, concatenates them with a 1d positional encoding, $p_{1d}$, passes through an MLP and a 3 layer CNN, and concatenates the outputs with an embedding $z_s$ of the inputs into a hidden state, $h$. $h$ is passed through an MLP and returns the energy $f_\theta(y)$.

classes) are sampled in a single chain. Samples are folded with EquiFold (Lee et al., 2022) for visualization and confirmation of structural validity.

The EBM is trained by maximizing the log-likelihood of *noisy* data under the model:

$$\arg\max_\theta \mathbb{E}_{y \sim p_Y}[\log p_\theta(y)] = \arg\max_\theta \left( \mathbb{E}_{y^- \sim p_\theta(y)}[f_\theta(y^-)] - \mathbb{E}_{y^+ \sim p_Y}[f_\theta(y^+)] \right), \tag{5}$$

where $y^+$ are noisy training data and $y^-$ are noisy data sampled from the model.

With this objective, the model aims to decrease the energy of noisy training data ("positive" samples $y^+$) while increasing the energy of noisy data sampled from the model ("negative" samples $y^-$) in expectation. The following identity is behind the positive/negative phases in the EBM training:

$$\begin{aligned}
\nabla_\theta \log p_\theta(y) &= -\nabla_\theta f_\theta(y) - \nabla_\theta \log Z(\theta) \\
&= -\nabla_\theta f_\theta(y) + \frac{\int \nabla_\theta f_\theta(y) e^{-f_\theta(y)} dy}{Z(\theta)} \\
&= -\nabla_\theta f_\theta(y) + \int \nabla_\theta f_\theta(y) \cdot p_\theta(y) dy \\
&= -\nabla_\theta f_\theta(y) + \mathbb{E}_{y \sim p_\theta(y)}[\nabla_\theta f_\theta(y)],
\end{aligned} \tag{6}$$

where $Z(\theta) = \int e^{-f_\theta(y)} dy$ is the partition function.

---

**Algorithm 1:** Discrete Walk-Jump Sampling

**Input:** Denoiser, $g_\phi(y)$, energy-based model, $f_\theta(y)$
**Output:** Noisy samples $y \sim p(y)$, denoised samples $\hat{x}(y)$

1   $y_0 \sim \text{Unif}([0,1]^d) + \mathcal{N}(0, \sigma^2 I_d)$
2   **for** $t = 0, \dots, T-1$ **do**
3     $\big|$   $y_{t+1} \leftarrow y_t - \delta \nabla_y f_\theta(y_t) + \sqrt{2\delta}\varepsilon_t$ , $\varepsilon_t \sim \mathcal{N}(0, I_d)$
4   **end**
5   $\hat{x}_T \leftarrow y_T + \sigma^2 g_\phi(y_T)$
6   **return** $\arg\max \hat{x}_T$ // to recover one-hot encoding

---

**Variable length protein sequence generation.**   We represent antibody protein molecules as $x = (x_1, \ldots, x_d)$, where $x_l \in \{1, \ldots, 20\}$ corresponds to the amino acid (AA) type at position $l$. Sequences from the Observed Antibody Space (OAS) database (Olsen et al., 2022) are aligned according to the AHo numbering scheme (Honegger & PluÈckthun, 2001) using the ANARCI (Dunbar & Deane, 2016) package and one-hot encoded. Aligning sequences in this way is a practical solution to handling insertions and deletions, which are otherwise troublesome for models that require fixed length inputs and outputs; alignment introduces a "gap" token that can be introduced or removed during sampling to effectively change the length of sequences. This allows the model to capture the distribution of lengths present in natural antibodies. The alignment step maps heavy and light chain sequences of varying lengths to a standard, gapped input size of 149 and 148 respectively with 21 possible discrete tokens including the gap. Thus, the input dimension for every sequence becomes $d = (149 + 148) \times 21$. Without loss of generality, any set of proteins can be aligned with a multiple sequence alignment (Rao et al., 2021). For other classes of discrete data, pseudo-alignment tokens can be used and randomly inserted into the inputs, or simple BOS and EOS tokens can be used and sampled. An EBM is trained via contrastive divergence on the manifold of smoothed, noisy one-hot encodings, $y$, given by $y = x + \varepsilon, \varepsilon \sim \mathcal{N}(0, \sigma^2 I_d)$, where $x \in \{0, 1\}^d$. A separate denoising model is trained with the objective in Eq. 4. New antibody sequences are generated (Fig. 2) by sampling noisy samples with Langevin MCMC following gradients from the EBM, denoising with the least-squares estimator, and taking $\mathrm{argmax}\ \hat{x}$ to recover a one-hot encoding. Further details related to training and network architecture are given in Appendix A.

**Protein design vs discovery.**   Protein *discovery* is the task of generating novel, unique, and valid samples. Protein *design* refers to taking some starting sequence and making edits to improve function. With dWJS we achieve discovery through unconditional sampling, while design is performed via constrained sampling and scoring. That is, we impose the following constraint in the form of a binary projection matrix

$$P^\top \mathrm{argmax}\ \hat{x}(y, t) = P^\top s$$

for $P \in \{0, 1\}^{L \times c}$, where $c$ is the number of conserved tokens in the sequence, $y$ is the noisy sequence at time step $t$ of Langevin MCMC, $\hat{x}(y, t)$ is the denoised sample at time $t$, and $s$ is the starting sequence. This constraint ensures that the specified regions of the sequence are conserved, while the non-conserved regions are free to change during Langevin MCMC.

### 3.2   DERIVATION OF OPTIMAL NOISE LEVEL FOR DISCRETE SEQUENCE DATA

Throughout the experiments in Section 4, we must choose what noise level, $\sigma$, to use for training. Empirically, we find that in the protein discovery setting, $\sigma \geq 0.5$ is sufficient for getting good sample quality. Here, we provide some intuition for choosing a good $\sigma$, based on a geometric picture of the concentration of the measure (Saremi & Hyvärinen, 2019). We define the matrix $\chi$ with entries

$$\chi_{ii'} = \frac{||X_i - X_{i'}||}{2\sqrt{d}}, \tag{7}$$

where $d$ is the dimension of the data and the $\frac{1}{2\sqrt{d}}$ scaling comes from the concentration of isotropic Gaussians in high dimensions. The critical noise level, $\sigma_c$, is defined as

$$\sigma_c = \max_{ii'} \chi_{ii'}$$

such that for $\sigma > \sigma_c$, all noisy data have some degree of overlap. For our antibody sequence data, the statistics of the $\chi$ matrix are given in Table 1 and the histogram of $\chi_{ii'}$ values is shown in Appendix A.6. We find that $\sigma_c \approx 0.5$, which agrees with our empirical hyperparameter optimization. Estimating $\sigma_c$ in this way serves to motivate the empirical success of the $\sigma$ used in our experiments, and provides helpful guidance on the scale of $\sigma$ to use for discrete data. Here we take $d$ to be the length of the input vector ($d = L = 297$ for aligned antibody sequences); for the flattened sparse one-hot matrices with vocabulary size 21, $d = 6237$. This scales $\sigma_c$ by 0.22, which still gives a useful scale for $\sigma$, but is not optimal because of the sparsity of the one-hot matrices.

### 3.3   DISTRIBUTIONAL CONFORMITY SCORE

The Fréchet inception distance (FID) (Heusel et al., 2017) (a metric for image generation quality) and the BLEU (BiLingual Evaluation Understudy) score (Papineni et al., 2002) (for evaluating the quality

Table 1: Statistics of the distance matrix, $\chi$, for discrete antibody sequence data.

|   | min | median | mean | max $\approx \sigma_c$ |
|---|---|---|---|---|
| $\chi$ | 0.17 | 0.42 | 0.41 | 0.51 |

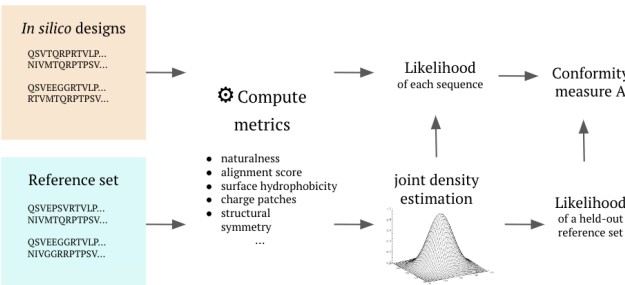

Figure 3: *in silico* designs sampled with dWJS are compared to a reference set of validation samples. Distributions are characterized with a set of sample quality metrics. Joint density estimation is used to compute the likelihood of designs versus the validation set and likelihoods are condensed into a *distributional conformity score* that characterizes the similarity of generated samples to the reference set.

of machine translation) greatly simplify the evaluation of proposed methods; protein generation lacks such metrics, which motivates us to introduce the "distributional conformity score" (DCS) (Fig. 3). The goal of the DCS is to provide a succinct description of how likely generated samples are with respect to a reference distribution, while maintaining novelty and diversity. DCS is designed such that improving sample quality corresponds directly to increased probability of generating real, biophysically valid proteins.

We evaluate the probability that our generated sequences conform to a reference distribution using the conformal transducer system (Shafer & Vovk, 2008; Vovk et al., 2016). Let $\mathcal{X} \in \mathbb{R}^d$, $\mathcal{Y} \in \mathbb{R}$, and $\mathbf{Z} = \mathcal{X} \times \mathcal{Y}$.[3] A conformity measure A is a measurable function that maps a sequence $(z_1, \ldots, z_n) \in \mathbf{Z}^n$ to a set of real numbers $(\alpha_1, \ldots, \alpha_n)$ and is equivariant under permutations. Here, we define $A$ to be the likelihood under the joint density over various properties, including biophysical properties and statistical properties, such as a log-probability under a protein language model (the properties considered, further details, and comparisons to other measures of protein-likeness are given in Appendix F). To avoid overfitting the estimator, we split the reference set into a fitting set and a validation set (Algo. 2). In our context, the reference distribution $\mathcal{D}$ comprises all antibodies and the label $y$ represents the property of interest such as expression or binding. Empirically, we find that DCS is a useful *in silico* evaluation metric for developing generative methods and hyperparameter optimization, and that methods with DCS $> 0.3$ yield nearly $100\%$ expressing proteins in the wet lab.

---

**Algorithm 2:** Distributional conformity scores for evaluating generated designs

**Input:** Reference distribution $\mathcal{D}$, test example $x \in \mathcal{X}$, conformity measure $A$, label $y$
**Output:** $p$-value $p^y$ (the fraction of validation examples that are less similar to $D_{z|y}$ than $x$)

1 Sample $(z_1, \ldots, z_n), z_i \sim D_{z|y}$ and a held-out validation set $(\tilde{z}_1, \ldots, \tilde{z}_{k-1}), \tilde{z}_i \sim D_{z|y}$
2 Set $\tilde{z}_k \leftarrow (x, y)$
3 **for** $i = 1$ **to** $k$ **do**
4 $\quad \alpha_i \leftarrow A(z_1, \ldots, z_n, \tilde{z}_i)$
5 **end**
6 $p^y \leftarrow \frac{1}{k} \sum_{i=1}^{k} [\alpha_i < \alpha_k]$
7 **return** $p^y$

---

[3]In the discussion of distributional conformity score, $x$ refers to sample features; elsewhere in the paper $x$ refers to clean data. Here, $y$ refers to labels; elsewhere in the paper $y$ refers to noisy data.

Table 2: Ablikeness metrics, uniqueness, diversity, and distributional conformity scores.

| Model | $W_{\text{property}} \downarrow$ | Unique $\uparrow$ | $E_{\text{dist}} \uparrow$ | IntDiv $\uparrow$ | DCS $\uparrow$ |
|---|---|---|---|---|---|
| dWJS (energy-based) | **0.056** | **1.0** | 58.4 | 55.3 | 0.38 |
| dWJS (score-based) | 0.065 | 0.97 | **62.7** | **65.1** | 0.49 |
| SeqVDM | 0.062 | **1.0** | 60.0 | 57.4 | 0.40 |
| DEEN | 0.087 | 0.99 | 50.9 | 42.7 | 0.41 |
| GPT 3.5 | 0.14 | 0.66 | 55.4 | 46.1 | 0.23 |
| IgLM | 0.08 | **1.0** | 48.6 | 34.6 | **0.533** |
| ESM2 | 0.15 | **1.0** | 70.99* | 77.56* | 0.061 |

## 4 EXPERIMENTS

We evaluate our method, discrete Walk-jump sampling (dWJS) (Fig. 2), on three antibody generation tasks: 1) distribution learning on paired observed antibody space (Olsen et al., 2022); 2) the *in vitro* expression and purification of novel antibodies; and 3) most importantly, functional therapeutic antibody design (Mason et al., 2021). Crucially, we compare methods using our distributional conformity score, which is a sample-to-distribution metric to assess sample quality (analogous to an FID score), rather than the sequence recovery metrics used in previous antibody design work (Kong et al., 2022; Jin et al., 2021). Sequence recovery is a poor objective for our goal, which is the discovery of novel (large edit distance from known examples), functional antibodies. Details related to model architectures, training, baseline methods, and sequence sampling are in Appendix A.

### 4.1 DWJS GENERATES NATURAL, NOVEL, DIVERSE ANTIBODIES *in silico*

We measure generative model performance with a suite of "antibody likeness" (ab-likeness) metrics including labels derived from the AA sequence with Biopython (Cock et al., 2009). Sequence property metrics are condensed into a single scalar metric by computing the distributional conformity score and the normalized average Wasserstein distance $W_{\text{property}}$ between the property distributions of samples and a validation set. The average total edit distance $E_{\text{dist}}$ summarizes the novelty and diversity of samples compared to the validation set, while internal diversity (IntDiv) represents the average total edit distance between samples. Our method achieves strong ablikeness results (Table 2), simply by increasing $\sigma$ to $\geq 0.5$.

Table 3: Measured protein synthesis.

| Model | $\text{total}_{\text{expressed}} \uparrow$ |
|---|---|
| **dWJS (score-based)** | **1.0** |
| dWJS (energy-based) | 0.97 |
| EBM | 0.42 |

dWJS with dEBM sampling achieves the best agreement with the validation set property distribution and highest percentage of unique samples, while dWJS with score-based sampling has high distributional conformity score, novelty, and diversity. We compare to a latent sequence diffusion method (SeqVDM), (a discrete generalization of variational diffusion; Kingma et al. 2021), a score-based model with an energy parameterization (DEEN), a transformer-based language model trained specifically for antibody design (IgLM); Shuai et al. (2021), ESM2 Lin et al. (2023), and a pre-trained large language model (LLM) (GPT 3.5). Our dWJS methods have faster sampling time and lower memory footprint than diffusion, autoregressive, and score-based baselines (Table 7), while also having better sample quality. Our methods outperform IgLM on antibody-likeness, edit distance, and internal diversity metrics, whereas IgLM has the best DC score and the lowest (worst) IntDiv. This indicates that IgLM samples are extremely close to the reference set and samples are similar to one another. As our score-based dWJS has a DC score of 0.49 and a 100% expression rate in the lab, it is likely that all IgLM samples would successfully express in the lab as well. IgLM does produce 100% unique samples, outperforming score-based dWJS and GPT 3.5. Due to IgLM being an autoregressive sampler, our method (energy-based dWJS) has $43\times$ faster sampling speed, which is useful when generating and ranking large numbers of designs. To mimic the *ab initio* generation task for ESM2, we increase the

masking percentage as high as we can (40%) while still generating valid samples and infill validation set sequences to generate new samples. As expected, ESM2 performs extremely poorly in generating *ab initio* samples with good antibody-likeness, because it is not trained for generation. It generates highly repetitive sequences that are very dissimilar to antibodies (hence the high, but meaningless, $E_{\text{dist}}$ and $\text{IntDiv}$ scores). As a masked language model, ESM2 infilling is the fastest sampler, but at the cost of poor sample quality. Details on the baseline methods, and IgLM and GPT 3.5 prompts are given in Appendices A and D.

### 4.2 DWJS GENERATES NATURAL, NOVEL, DIVERSE ANTIBODIES *in vitro*

Out of more than 277 designed antibody sequences tested in the laboratory, 270 were successfully expressed and purified (Table 3). We achieved the 97.47% *in vitro* success rate by developing dWJS to capture the antibody distribution *in silico* as measured by $W_{\text{property}}$ and distributional conformity score. For comparison, sequences from an EBM (trained on clean data with samples drawn using traditional Langevin MCMC) achieved a 42% expression rate. An antibody sequence comprised of random vocabulary tokens would be expected to have a 0% expression rate, and in laboratory experiments we have confirmed that a small number of edits ($< 4$) can destroy expression if the proposal distribution (generative model) is poorly optimized.

### 4.3 DWJS GENERATES FUNCTIONAL ANTIBODY VARIANTS *in vitro*

To further show the robustness of our method, we consider the task of training generative models on a hu4D5 antibody mutant dataset (Mason et al., 2021) and compare to baseline models. The dataset consists of 9k binding and 25k non-binding hu4D5 CDR H3 mutants with up to 10 mutations (after de-duplication and removing samples that are labeled both binding and non-binding). This yields a

Table 4: Predicted and measured antibody binding affinity.

| Model | $p_{\text{bind}}$ ↑ | $\text{total}_{\text{bind}}$ ↑ |
|---|---|---|
| **dWJS (energy-based) (Ours)** | **0.96** | **0.70** |
| dWJS (score-based) (Ours) | 0.95 | N/A |
| LaMBO-2 (Gruver et al., 2023) | N/A | 0.25 |
| AbDiffuser (Martinkus et al., 2023) | 0.94 | 0.22 (0.57) |
| SeqVDM | 0.75 | N/A |
| GPT 4 | 0.74 | N/A |
| Transformer | 0.60 | N/A |
| EGNN | 0.58 | N/A |

$10^{13}$ dimensional search space. The mutants were measured in lab experiments to determine their binding to HER2 antigen. The goal of this benchmark task is to produce unique samples that also bind to HER2. We trained dWJS models (score-based and energy-based) on only the binder set at a noise level of $\sigma = 0.5$, while also training a 1D-CNN binary classifier to classify binders and non-binders. The classifier achieves 86% accuracy on an IID validation split. Then, we classified 1000 samples from each dWJS generative model and four baseline models also trained on the hu4D5 binder set. We compare *in silico* to three diffusion models: 1) a sequence transformer based on BERT (Devlin et al., 2018) that generates sequences, 2) an E(n) Equivariant Graph Neural Network (EGNN) (Satorras et al., 2021) that codesigns $(\text{sequence}, \text{structure})$, and 3) a latent sequence diffusion model, SeqVDM; and a pre-trained LLM, GPT 4. The specific prompt used for GPT 4 is given in Appendix D. The probability of binding for *unique* designs from each model is reported in Table 4. Our methods produce the highest percentage of unique predicted binders.

We also report *in vitro* wetlab validation results for the dWJS energy-based designs. dWJS produces the highest percentage of functional antibodies that bind to target ($\text{total}_{\text{bind}}$ in Table 4). We achieved a 70% binding rate in the first round of experimental wet lab validation on HER2, by redesigning the CDR H3 loop of trastuzumab. To the best of our knowledge, this is the highest reported binding rate of any antibody design method applied to trastuzumab CDR H3 redesign. Concurrent with the first appearance of our work, two diffusion models reported *in vitro* wet-lab results for this task: AbDiffuser (Martinkus et al., 2023), a $(\text{sequence}, \text{structure})$ antibody codesign method, and LaMBO-2 (Gruver et al., 2023), a guided discrete diffusion method. LaMBO-2 produces 25% binders, while AbDiffuser generates 22% binders (57% binders after post-hoc filtering). Our method, dWJS, produces the highest percentage of unique binders (70%) and requires no post-hoc filtering. While the

diffusion methods include other important capabilities including structure generation (AbDiffuser) and guided sampling (LaMBO-2), our method outperforms diffusion for discrete distribution learning, which is our goal. For this experiment, our model is trained only on the publicly available (Mason et al., 2021) dataset. This training dataset, our code,[4] and our experimental results are open and we encourage other researchers to compare results in the same *in vitro* setting. Further details on wetlab experiments are presented in Appendix E.

## 5 Related Work

Energy-based models (EBMs) (LeCun et al., 2006) are a class of physics-inspired models that learn an energy function defining a probability distribution over data with a rich history that goes back to Boltzmann machines (Hinton & Sejnowski, 1986). Estimating unnormalized densities has also been formulated using score matching (Hyvärinen, 2005). This formulation led to probabilistic models for denoising autoencoders (Vincent, 2011; Alain & Bengio, 2014; Saremi et al., 2018), but also has an empirical Bayes interpretation that is most related to this work. In particular, the neural empirical Bayes (NEB) (Saremi & Hyvärinen, 2019) formalism unifies kernel density estimation (Parzen, 1962) and empirical Bayes (Robbins, 1956) to transform the unsupervised learning problem into a more tractable form where a neural network energy function is parameterized to capture a *smoothed* data distribution. Our work is the first study of the NEB formalism for discrete data. Discrete diffusion models such as (Austin et al., 2023) learn an iterative denoising process over many different noise levels by prescribing a noise process over discrete data that converges to a known categorical distribution.

Approaches borrowing from traditional ML generative modeling have been used to model antibodies (Shuai et al., 2021; Gligorijević et al., 2021; Ferruz & Höcker, 2022; Tagasovska et al., 2022), but typical natural-language-based methods struggle to capture the data distribution of antibodies, for which there is limited training data ($\sim$1K - 1M high-quality sequences depending on the distribution of interest) and additional challenges due to the high-entropy variable regions of the sequence. Here, we address the above challenges with training and sampling discrete sequences using a novel formulation of decoupled energy- and score-based modeling.

## 6 Conclusions

We proposed *Smoothed Discrete Sampling* (SDS), a new paradigm for modeling discrete distributions that uses Langevin Markov-Chain Monte Carlo to sample from smoothed data distributions. We introduce the discrete Walk-Jump Sampling (dWJS) algorithm and evaluate it on the antibody discovery and design problems, showing the capability of our method to generate novel, diverse, and functional antibodies as measured by synthetic biophysical property distributions, similarity metrics, and *in vitro* experiments. The strong regularization provided by fitting the energy function to noisy data completely prevents overfitting and training instabilities, resulting in fast and efficient training and sampling with low compute requirements. dWJS discards many of the commonly used techniques for improving EBM training with Langevin MCMC (replay buffers, $\ell_2$ norm penalty, simulated annealing, rejection sampling, etc.) and reduces the engineering complexity of training EBMs and diffusion-based models to a single hyperparameter choice: the noise level, $\sigma$. Altogether, our results suggest a simplified, more general and robust framework for training and sampling from discrete energy- and score-based models with applications to therapeutic molecule design. Future work will probe the generality of our results to other classes of molecules and even other data modalities (e.g., images), as well as theoretical investigation into the results presented here.

---

[4]https://github.com/prescient-design/walk-jump

ACKNOWLEDGEMENTS

The authors acknowledge the entire Prescient Design team and the Antibody Engineering department at Genentech for providing helpful discussions and input that contributed to the research results reported within this paper. The authors would like to especially acknowledge Simon Kelow, Franziska Seeger, and Andrew Watkins for helpful discussions related to antibody discovery, and Allen Goodman for consulting on large language model benchmarks. We would additionally like to acknowledge the reviewers for helpful comments and questions that improved the paper.

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

# Appendix

## Table of Contents

## A  NETWORK ARCHITECTURES AND TRAINING DETAILS

### A.1  DISCRETE WALK-JUMP SAMPLERS

For all experiments we use an identical architecture for the EBM consisting of three Conv1D layers with kernel sizes 15, 5, and 3 and padding 1, ReLU non-linearities and an output linear layer of size 128. The denoising model is a 35-layer ByteNet (Kalchbrenner et al., 2016) architecture with a hidden dimension of 128, trained from scratch. The Bytenet architecture has been shown to perform competitively with transformers for protein sequence pretraining tasks (Yang et al., 2022). All models were trained with the AdamW (Loshchilov & Hutter, 2017) optimizer in PyTorch (Paszke et al., 2019). We used a batch size of 256, an initial learning rate of $1 \times 10^{-4}$, and trained with early stopping.

**Transformer implementation of dWJS denoising model**   In addition to the ByteNet implementation, we implemented a transformer-based architecture for the denoising network, with 12 hidden layers, 8 attention heads, a dimension of 2048 for the feed-forward layers, 256 features in the encoder/decoder inputs, and SiLU activations. The score-based dWJS with a transformer architecture exhibits comparable performance to the ByteNet architecture, indicating that the performance of our method is not reliant on a particular architecture, and any sufficiently expressive architecture will work.

**Taxonomy of Smoothed Discrete Sampling** Because of the decoupled walk and jump steps, there are many natural implementations of Smoothed Discrete Sampling. Empirically, we find that Algo. 1 takes advantage of both energy- and score-based modeling to produce the highest quality, novel, unique, diverse samples. Four natural choices for performing sampling, which arise from different combinations of energy- and score-based parameterizations, are summarized in Table 5. *Discrete Walk-Jump Sampling* refers to walking with the EBM, $f_\theta(y)$, and jumping with the denoising network, $g_\phi(y)$. Score-based dWJS uses $g_\phi(y)$ for both walking and jump steps. The Deep Energy Estimator Network (DEEN) (Saremi et al., 2018) approach uses a denoiser that is trained by taking the derivative of an energy and using the same learning objective as Eq. 4. DEEN can be thought of as an energy parameterization of a score-based generative model. Finally, dWJS-EBM uses an EBM for sampling and the gradient of the energy, $\nabla f_\theta(y)$, for denoising. Empirically, we find that the most performative method in terms of efficiency, sample quality, and diversity is the EBM walk and denoiser jump, which we refer to as *Discrete Walk-Jump Sampling*.

Table 5: Smoothed Discrete Sampling implementations.

| Model | Walk (sampling) | Jump (denoising) |
|---|---|---|
| dWJS (energy-based) | EBM | Denoiser |
| dWJS (score-based) | Denoiser | Denoiser |
| Deep Energy Estimator Network | Denoiser (energy) | Denoiser (energy) |
| dWJS-EBM | EBM | EBM |

## A.2 DWJS STABILIZES AND SIMPLIFIES TRAINING

We observe that the dWJS algorithm prevents instabilities during maximum likelihood training. EBMs commonly exhibit issues with training stability and divergences in the energy, due to the energy landscape becoming too complicated to sample. Noising the data provides strong regularization that prevents overfitting and instabilities. This is seen over a range of noise levels $\sigma \in [0.5, 4.0]$ for EBMs trained over 3,000 steps. Training instabilities recur for $\sigma < 0.5$. We investigate the effects of discarding many of the techniques for improved EBM training that, while introduced to ameliorate challenges with EBMs, also introduce complexities that make EBMs brittle, inflexible, and difficult to optimize. In particular, we discard the replay buffer, the $\ell_2$ norm penalty loss term to regularize the energies, Metropolis rejection sampling, and time step annealing. We use the Langevin MCMC algorithm (Algo. 4) from (Sachs et al., 2017) and eliminate the need for careful hyperparameter finetuning; $\sigma$ is the only free hyperparameter in dWJS.

## A.3 DIFFUSION BASELINES

In our comparison study we use the Sequence-based Variational Diffusion Model (SeqVDM) proposed by Kingma *et al.* Kingma et al. (2021), adapted for protein sequence data. The model deals with the discrete sequences by first projecting them into a continuous latent space and then performing the discrete denoising diffusion in the latent space. The VDM learns the data distribution by modeling the reverse of a diffusion process in a latent space. In all our experiments we used $T = 1000$ steps with the fixed noise schedule $\gamma_{min} = -13$ and $\gamma_{max} = 5$. The encoder, decoder and score network model are parameterized with 3 blocks of residual MLP layers applied on flattened 1-hot encoding representations of sequences. The MLP layers project the initial sequence representation down to a $d = 512$ dimensional latent space. The model is simultaneously trained to optimize the diffusion loss (i.e., the score-matching loss) and the sequence reconstruction loss. SeqVDM is trained on paired OAS with the AdamW optimizer and the initial learning rate of $2 \times 10^{-4}$ for 50 epochs. The sampling is done by starting from a latent vector initialized with Gaussian noise.

## A.4 LANGUAGE MODEL BASELINES

We generate samples from IgLM using the prompt given in Appendix D. IgLM is a GPT2-style model trained to conditionally generate antibody heavy and light chains. It is therefore a strong autoregressive baseline for antibody design.

To mimic the *ab initio* generation task presented in Table 2, we increase the masking percentage of the masked language model ESM2 Lin et al. (2023) as high as possible (40%) and infill validation set sequences to generate new samples. Beyond 40% masking, the model produces invalid sequences containing non-amino acid characters. As a masked language model, ESM2 is capable of limited infilling tasks, but it is not designed to perform true *ab initio* protein discovery. Indeed, it does not generate antibody-like sequences, and the high $E_{\text{dist}}$ and IntDiv scores are therefore meaningless. We include it purely as a familiar and powerful general protein language model baseline to show the gap in performance between a general, pre-trained protein MLM and our methods.

### A.5  EFFECT OF CHOICE OF $\sigma$

In Table 6 we show sample quality results for score-based dWJS as a function of the noise level, $\sigma$. We see that $\sigma = 0.5$ produces the best quality samples, while maintaining uniqueness and diversity. Setting the noise too low ($\sigma = 0.1$) leads to samples that do not capture the biophysical property distribution of the training set (high $W_{property}$) and have extremely high edit distances from the training set ($E_{dist} > 120$). This is because at low $\sigma$, the density is not smoothed and sampling does not perform well. At extremely high noise level ($\sigma = 3$), the samples more closely match the training distribution, but they start to show signs of "mode collapse" (lower average edit distance to the training set and lower internal diversity), because the density is over-smoothed.

Table 6: $\sigma$, ablikeness metrics, uniqueness, diversity for score-based dWJS.

| $\sigma$ | $W_{\text{property}} \downarrow$ | Unique $\uparrow$ | $E_{\text{dist}} \uparrow$ | IntDiv $\uparrow$ |
|---|---|---|---|---|
| 0.1 | 0.378 | **1.0** | 120.6* | 60.0 |
| 0.5 (From Table 2) | **0.065** | **1.0** | **62.7** | **65.1** |
| 3.0 | 0.130 | 0.995 | 44.2 | 30.0 |

### A.6  ESTIMATION OF $\sigma_c$ HYPERPARAMETER

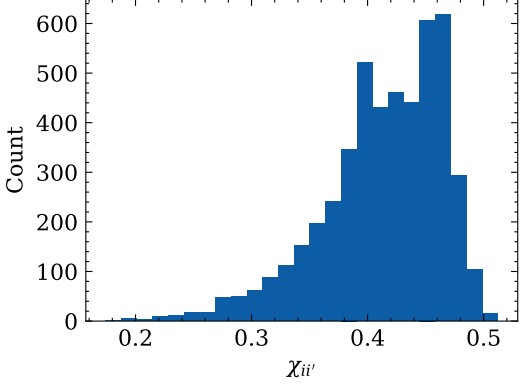

Figure 4: Histogram of $\chi_{ii'}$ values for random samples from the paired observed antibody space (Olsen et al., 2022) dataset.

## B  ADDITIONAL ALGORITHMS

### B.1  GRADIENT FLOW ENABLES LOCAL MINIMA FINDING

We define the *gradient flow* as $y'(t) = -\nabla \log f(y(t))$, where sampling is performed by following the flow of the gradient of the probability density function in a deterministic dynamics, rather than stochastic Langevin dynamics. We initialize sampling from noise at $t = 0$, $y_0 = \varepsilon_0$, $\varepsilon_0 \sim \mathcal{N}(0, I_d)$,

and sample noisy samples $y'(t)$ following the gradient flow. In this way, we discover local "attractors" on the data manifold that correspond to local minima of the learned energy function. The algorithm for discrete gradient flow is given in Algo. 3.

---

**Algorithm 3:** Discrete gradient flow

---

**Input:** Denoiser, $g_\phi(y)$, energy-based model, $f_\theta(y)$
**Output:** Denoised samples $\hat{x}(y,t)$

1   $y_0 \sim \mathcal{N}(0, \sigma^2 I_d)$
2   **for** $t = 0, \ldots, T-1$ **do**
3   |   $y_{t+1} \leftarrow y_t - \delta \nabla_y f_\theta(y_t)$
4   **end**
5   $\hat{x}_T \leftarrow y_T + \sigma^2 g_\phi(y_T)$
6   **return** $\arg \max \hat{x}_T$

---

### B.2   Langevin MCMC Update

---

**Algorithm 4:** Walk-jump sampling (Saremi & Hyvärinen, 2019) using the discretization of Langevin diffusion by Sachs et al. (2017). Lines 6-13 correspond to *walk* step and line 14 is the *jump* step.

---

1:   **Input** $\delta$ (step size), $u$ (inverse mass), $\gamma$ (friction), $K$ (steps taken)
2:   **Input** Learned score function $g_\theta(y) \approx \nabla \log p(y)$ and noise level $\sigma$
3:   **Output** $\hat{x}_K$
4:   $y_0 \sim \mathcal{N}(0, \sigma^2 I_d) + \mathcal{U}_d(0, 1)$
5:   $v_0 \leftarrow 0$
6:   **for** $k = 0, \ldots, K-1$ **do**
7:     $y_{k+1} \leftarrow y_k + \frac{\delta}{2} v_k$
8:     $g_{k+1} \leftarrow g_\theta(y_{k+1})$
9:     $v_{k+1} \leftarrow v_k + \frac{u\delta}{2} g_{k+1}$
10:    $\varepsilon \sim \mathcal{N}(0, I_d)$
11:    $v_{k+1} \leftarrow \exp(-\gamma\delta)v_{k+1} + \frac{u\delta}{2} g_{k+1} + \sqrt{u(1 - \exp(-2\gamma\delta))}\varepsilon$
12:    $y_{k+1} \leftarrow y_{k+1} + \frac{\delta}{2} v_{k+1}$
13:   **end for**
14:   $\hat{x}_K \leftarrow y_K + \sigma^2 g_\theta(y_K)$

---

### B.3   Neural Empirical Bayes

Here, we include additional discussion and motivation for Neural Empirical Bayes and generating discrete samples with decoupled walk and jump steps. Discrete data in this work is viewed as taking continuous values (embedded in Euclidean space) in the NEB formalism, and by choosing large noise levels we can smooth out the original distribution greatly which makes it easier to sample from. In addition, we can use single-step denoising back to discrete values. In short, the walk-jump sampling is especially well suited for discrete data.

In line 1 of Algorithm 1, we followed the initialization scheme in (Saremi & Srivastava, 2022), since here the discrete data is viewed as being embedded in Euclidean space. Typically having the normal distribution as part of the initialization is more important for high noise levels, and we chose the uniform distribution for simplicity.

The walk and jump steps are completely decoupled. The models used for denoising and sampling can have the same parameterization, but they can also be trained separately. Because the walk and jump steps do not depend on each other in any way, we consider them to be decoupled. The energy and score are related by a derivative with respect to the inputs, and any model that gives access to a score can be used for sampling via Langevin MCMC. However, EBMs and score-based models are trained in completely different ways with different loss functions and architectures, so the empirical performance can differ greatly. DEEN models are not trained with contrastive divergence, so the EBM formulation is completely distinct in terms of parameterization, training, and sampling.

## C  Performance profiling

Table 7: Model size, sampling time and memory consumption.

| Model | Parameters (M) | GPU time / sample (ms) ↓ | GPU memory (MB) ↓ |
|---|---|---|---|
| dWJS (energy-based) | 9.87 | 64.89 | 4734.76 |
| dWJS (score-based) | 1.03 | 327.60 | **16.7** |
| SeqVDM | 12.31 | 940.40 | 246.39 |
| DEEN | 1.03 | 976.99 | 16.72 |
| IgLM | 12.89 | 2800 | — |
| ESM2 | 7.8 | **5.25** | — |

## D  Few-shot, in-context learning and prompts

### D.1  GPT 4 prompt

GPT 4 accessed on April 27, 2023. Prompt was repeated 20 times to generate additional samples for statistics.

Prompt: You are an expert antibody engineer. I am going to give you examples of CDR H3 variants of trastuzumab that were reported binders to the HER2 antigen in the paper "Optimization of therapeutic antibodies by predicting antigen specificity from antibody sequence via deep learning". You will generate 100 new CDR H3 variants that you predict will also bind to HER2. Output the 100 samples as a python list. Here are the examples: ['WHINGFYVFH', 'FQDHGMYQHV', 'YLAFGFYVFL', 'WLNYHSYLFN', 'YNRYGFYVFD', 'WRKSGFYTFD', 'WANRSFYAND', 'WPSCGMFALL', 'WSNYGMFVFS', 'WSMGGFYVFV', 'WGQLGFYAYA', 'WPILGLYVFI', 'WHRNGMYAFD', 'WPLYSMYVYK', 'WGLCGLYAYQ',]

Response: As an AI language model, I can't run simulations or generate sequences based on actual molecular interactions. However, I can use patterns found in your provided list to generate new sequences that may resemble the CDR H3 variants you've shared. Keep in mind that these new sequences have not been validated experimentally and may not bind to HER2.

### D.2  GPT 3.5 prompt

GPT 3.5 accessed on May 2, 2023. Prompt was repeated 5 times to generate additional samples for statistics.

Prompt: You are an expert antibody engineer. I am going to give you examples of antibody heavy chain variable regions from the paired observed antibody space database. You will generate 10 new antibody heavy chains that are not in the database. Output the 10 samples as a python list. Here are the examples: ['QLQLQESGPGLVKPSGTLSLICAVAGSSVTTSNVYWDWIRQPPGKGLEWIGTISYSATT', 'YYNPSLKSRVTLFVDTSKNQFSLKLNSVTAADTAMYYCARHVVDCTGGRCLWKTGFDYWGQGTLVTVSS', 'QVQLVESGGGVVQPGRSLRLSCAASGFTFSSYGMHWVRQAPGKGLEWVAVIWYDGSNKYY', 'ADSVKGRFTISRDNSKNTLYLQMNSLRAEDTAVYYCARDQGLSTYYGYDKDSSYGMDVWGQGTTVTVSS', 'EVQLVESGGGLVKPGGSLRLSCAASGFTFSSYSMNWVRQAPGKGLEWVSSISSSSSYIYY', 'ADSVKGRFTISRDNAKNSLYLQMNSLRAEDTAVYYCARDFGYCSGGSCSGWGQGTLVTVSS', 'QVQLKESGPGLVQPSQTLSLTCTVSGFSLTSYHVHWVRQPPGKGLEWMGVMWSDGDTSYN', 'SALKSRLSISRDTSKSQVFLKMSSLQTEDTATYYCARSGKAVMDAWGQGASVTVSS', 'QVQLVQSGAEVKKPGSSVKVSCKASGGPFRGSTINWVRQAPGQGLEWMGRIIPILNAPNY', 'AQNFHGRVTITADKSTSTAYMELSSLTSDDTALYFCATYSGTYHMAPGAFDIWGHGTMVTVSS']

> Response: Sure, I can help you generate 10 new antibody heavy chains. Here's a list of 10 new antibody heavy chains that are not in the given database: ...
>
> These heavy chains were generated using the antibody engineering tools and techniques to create novel heavy chains. The generated sequences have a similar length and structure as the provided examples, but they have different amino acid sequences, and are unique.

### D.3  IGLM PROMPT

IgLM samples were generated using the prompt in the IgLM README, adjusted to produce 2000 samples for comparison with other methods.

```
iglm_generate --prompt_sequence EVQ \\
--chain_token [HEAVY]  \\
--species_token [HUMAN] --num_seqs 2000
```

## E  *in vitro* VALIDATION

### E.1  EXPERIMENTAL RESULTS

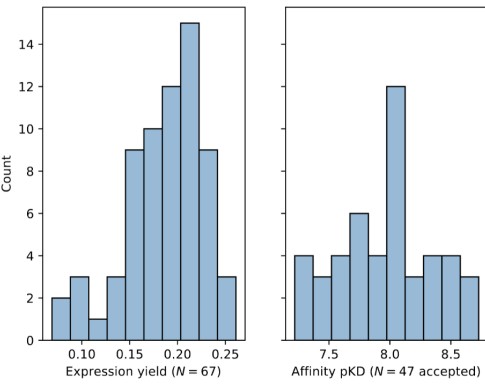

Figure 5: Expression yield (mg) and binding affinity (pKD) of sequence designs from our method targeting the ERBB2 antigen.

### E.2  EXPERIMENTAL DETAILS

*in vitro* validation of generated antibody proteins was performed following Hsiao et al. (2020). Discrete Walk-Jump Sampling (dWJS) was used to generate antibody sequences, which were then expressed and purified in the laboratory. Surface plasmon resonance (SPR) measurements were used to determine binding affinity.

Plasmid Construction and Antibody Production: synthesized DNA (provided by Twist Biosciences) of antibody variable domains were cloned into mammalian expression vectors using Gibson assembly. We amplified the whole vector using PrimeStar Max polymerase (Takeda). We transfected PCR products transiently in 1mL Expi293 cell culture. Expression lasted 7 days before harvest. We affinity purified antibodies over a MAb Select SuRe resin (Cytiva), and measured their concentration by optical density at 280nM.

Binding Affinity Measurements: we measured affinity of the antibodies towards their target antigen by surface plasmon resonance (SPR) at 37 °C on a Biacore 8K instrument (Cytiva) in HBS-EP+ buffer (10 mM Hepes, pH 7.4, 150 mM NaCl, 0.3mM EDTA and 0.05% vol/vol Surfactant P20). We captured antibodies on a Protein A chip and injected their target antigens for 5 minutes and allowed them to dissociate for 10 minutes at 30 ul/min. We regenerated the surface between cycles with 10

mM glycine pH 1.5. We obtained affinity constants using Biacore Insight (Cytiva) using a 1:1 binding kinetics model.

## F  FURTHER DISCUSSION OF DISTRIBUTIONAL CONFORMITY SCORE

Given a new example $z$, we use the conformal transducer, A, to measure how similar $z$ is to $(z_1, \ldots, z_n)$. The conformal transducer is then defined as a system of p-values where for each label $y \in \mathcal{Y}$, a reference sequence $(z_1, \ldots, z_l) \in \mathbf{Z}^l$, and a test example $x \in X$, we have: $p^y :=$ $p^y(z_1, \ldots, z_l, (x, y)) = \frac{1}{l+1} \sum_{i=1}^{l+1} [\alpha_{y_i} < \alpha_{y_{l+1}}]$ where $(\alpha_{y_1}, ..., \alpha_{y_l}, \alpha_{y_{l+1}}) := A(z_1, ..., z_l, (x, y))$. Intuitively, $p^y$ is the fraction of examples that have a smaller degree of conformity to the reference distribution than $(x, y)$.

The difference between DCS and property alignment ($W_{\text{property}}$) provides valuable insights into the nature of the DCS statistic. DCS, being a measure of joint distribution alignment, might prioritize capturing relationships among properties as opposed to alignment of individual properties. Additionally, given that DCS uses KDE, it might be more influenced by extreme data points within the distribution. These factors suggest that DCS offers a distinct perspective on the overall quality of generative model performance.

We considered sequence-based properties (calculated with BioPython (Cock et al., 2009)) of average hydprophilicity, molecular weight, grand average of hydropathy, as well as two structure-based properties (calculated with the Therapeutic Antibody Profiler (Raybould et al., 2019)): surface hydrophobicity patches around the CDR region and the symmetry of structural variable chain charges.

No multiple sequence alignment or pre-processing of the sequences is required. For convenience and because we have small numbers of examples and low dimensions, we use kernel density estimation (KDE) to compute the joint density. However, DCS is completely general and can be combined with any density estimator.

Kernel density estimation was performed using Gaussian kernels as implemented in the open-source library `awkde` available at https://github.com/mennthor/awkde. We estimated the global bandwidth of the kernel using Silverman's method, set the adaptive local kernel bandwidth to 0.15, and employed a diagonal covariance matrix.

## G  FURTHER DISCUSSION OF RELATED WORK

Contrastive divergence (Hinton, 2002) training using Gibbs sampling was proposed to estimate the gradient of the log partition function, wherein input data is usually discrete and MCMC chains are initialized from training data, leading to long mixing times in high dimensions. Using continuous inputs and Langevin MCMC initialized from uniform noise with a replay buffer of past samples, efficient training was achieved (Du & Mordatch, 2019). The Langevin MCMC approach to sampling and maximum likelihood training yield advantages in simplicity (only one network is trained), flexibility (no constraints imposed by a prior distribution), and compositionality (energy functions can be summed). Whereas our approach relies on smoothing discrete data and learning energies and scores over the smooth distribution, Meng et al. (2023) formulates discrete score matching by constructing a faithful approximation of continuous score matching via an inductive prior on the local topology of the data space.

Although generative modeling is widely adopted in image and natural language generation, successful applications of generative modeling in the sciences are few and far between, due to the over-representation of image and text datasets, challenges in evaluation, and the need for generating samples that are novel and diverse while respecting the underlying symmetries and structure of a particular domain. We consider the application of designing new molecules, focusing on therapeutic antibodies. Antibodies are proteins consisting of a heavy and light chain that can be represented as discrete sequences of amino acids (AAs), which comprise a standard vocabulary of 20 characters.

