# OpenReview forum: "Protein Discovery with Discrete Walk-Jump Sampling"
_ICLR.cc/2024/Conference — ICLR 2024 oral_

### Official Review · Reviewer_BFxy · 2023-10-26

**Soundness:** 4 excellent
**Presentation:** 4 excellent
**Contribution:** 4 excellent
**Rating:** 8
**Confidence:** 4

**Summary:**

The paper introduces an interesting, novel approach to sampling new, diverse antibody sequences and validates the method's performance using real wet-lab experiments with real antibodies. They explore a couple variations on energy-based models with some interesting tricks-of-the-trade such as the pretrained denoiser to help guide the sampling. The experimental results are strong.


==After author's response==
I appreciate all of the details in response to my question. I continue to advocate for acceptance of this high-quality paper. My only final feedback is that the discussion of mixing should mention that mixing is not a concern for autoregressive models, since for them it's trivial to draw independent samples.

**Strengths:**

The paper does real wet-lab experiments! By that, I don't mean just that experiments were performed in a wet lab in a toy setting, but instead that the authors did real-world antibody design.

The paper provides an interesting/refreshing take on how to sample antibodies that is quite different from most papers' use of language models. It's drawing on a whole line of work on energy-based models with details that are both old-school and new-school. There will be many ML+Bio researchers that will be unfamiliar with this background material and will benefit from seeing this paper.

I thought it was interesting that they included GPT3.5 as a baseline. Sometimes the results are surprisingly reasonable! As generalist foundation models get stronger, such a baseline seems important.

The authors have release code for their modeling, which is key because it is more niche than LM approaches for which there are standard packages.

**Weaknesses:**

Some of the comparisons to autoregressive language models (LMs) were a bit under-developed. For example, the paper mentions fast mixing sampling, but this is trivial for LMs. Second, LMs provide a natural way to condition on metadata; how would the current model do that? Finally, claims that LM sampling is expensive in terms of compute aren't that compelling, since for these antibody design applications the wet-lab experiments are orders of magnitude more expensive than the compute to generate the library.


The 'distributional conformity scores' (DCS) section was interesting and cool, but not central to the paper and felt like a way to add extra math into the paper. Please clarify: was this used to filter any of the samples for the wet-lab experiments or was it only used to provide the rightmost column in Table 2? Given that your proposed does not have strong DCS scores compared to some other baselines, how should I be interpreting these results? Overall, I would recommend removing the DCS content and allocating more space in the main paper to details/derivations of the model fitting/sampling.

**Questions:**

The idea of relaxing the discrete problem to a continuous modeling problem on one-hots is interesting. I was surprised, however, that you treated them as one-hots instead of elements of a probability simplex. Have you explored this? A key advantage is that you could sample from the probabilities rather than taking the argmax in the final step.

One of my concerns for using your method vs. LMs is that there are a number of difficult-to-set hyperparameters. Can you discuss the sensitivity to these choices? How did you validate them?

The DCS seems like a good way to filter samples. Is there a way that you could evaluate the impact of such filtering in-silico?

The DCS is a one-vs-rest comparison, using a single query vs. a reference set. How could you extend this to generate a *diverse* set of samples?

The paper claims informally that the sampling mixes quickly. Is there a way to demonstrate this quantitatively?

---

> ### Author Response · Authors · 2023-11-20
>
> We thank the reviewer for their enthusiastic and encouraging review of our work, and the emphasis on the novelty of our approach and our real-world in vitro experiments. Below we have responded to each point raised by the reviewer:
>
> 1. The reviewer raises a good point that autoregressive LMs have very different capabilities and advantages compared to non-autoregressive models. The natural way to condition on metadata for dWJS is to add prefix tokens for metadata (e.g., species) and sample with those tokens included. It is also true that for wet-lab integration, sampling speed/computational cost is negligible compared to lab experiments. The advantages in sampling speed are more interesting for iteration and method development of new ML methods, where fine-tuning LMs may be insufficient and pre-training from scratch may be impractical. Nevertheless, we believe that pre-trained LLMs are an under-studied model type in ML for biology, hence our investigations into their baseline performance for sequence generation.
>
> 2. DC Score is primarily a way to formulate a single, aggregated scalar objective based on feature engineering, to detect if a generative model is producing samples that are “in-distribution” with respect to the training data. We find that DCS is a reliable measure to indicate if samples will express in the lab, and we used DCS as an in silico metric for developing our methods, which allowed us to achieve a high expression rate in the first round of lab experiments. However, we find that for DCS > 0.3, almost all proteins are expressed. We have added this additional context in the discussion of DCS, to more clearly motivate its usage and explain how it was used in silico before conducting wet-lab experiments. We have clarified these points in the discussion of DC score by moving some of the technical details to Appendix F to respect the page limits and adding the following text to Section 3.2: “No multiple sequence alignment or pre-processing of the sequences is required. For convenience and because we have small numbers of examples and low dimensions, we use kernel density estimation (KDE) to compute the joint density. However, DCS is completely general and can be combined with any density estimator…​​Empirically, we find that DCS is a useful in silico evaluation metric for developing generative methods and hyperparameter optimization, and that methods with DCS > 0.3 yield nearly 100% expressing proteins in the wet lab.”
>
> 3. We have not explored treating the modeling problem over a probability simplex and sampling, primarily because “greedy” decoding with argmax worked so well for achieving high sample quality and diversity. Sampling from the probabilities is certainly an interesting future direction for increasing the control over sampling.
>
> 4. The most important hyperparameter to set is the noise level, sigma. In section 3.2 we derive a “critical noise level” of ~0.5 for our data, which agrees with our empirical findings that 0.5 < sigma < 1.0 leads to good sampling. To provide some intuition, sigma should be large enough to smooth the data and make sampling easier, but not so large that the denoising network cannot recover “valid” samples. The “default” value of sigma in [0.5, 1.0] works well. We have included additional experiments and discussion in Appendix A.5 and Table 6 to illustrate this point.
>
> 5. The DC score is an effective method for detecting samples that are not in-distribution with respect to the training data. Empirically, we find that DCS > 0.3 leads to samples that nearly all express in the lab. A low DC score (e.g., 0.06 for ESM2 shown in Table 2) indicates protein samples that will not express. We have updated the manuscript to clarify this point.
>
> 6. It is interesting to think about how DC score could be generalized to also account for diversity. In our current setup, we use DCS to evaluate conformity to a reference set, and compute additional edit-distance-based metrics to evaluate diversity. To evaluate diversity, it might be possible to treat a set of generated samples as the reference set and do leave-one-out evaluation of each design’s conformity to the rest of the designs.
>
> 7. Our observation that dWJS easily samples many antibody germlines (Figure 1) in a single MCMC chain shows the fast mixing time of our method - this would be impossible with a traditional energy-based model, and is not a capability of autoregressive models. In future work, we intend to do a more theory-focused exploration of the mixing time of dWJS compared to traditional MCMC methods.

---

### Official Review · Reviewer_RGZR · 2023-10-30

**Soundness:** 3 good
**Presentation:** 4 excellent
**Contribution:** 3 good
**Rating:** 8
**Confidence:** 3

**Summary:**

The authors present a new approach for modeling discrete data distributions called Smoothed Discrete Sampling (SDS), founded on the neural empirical Bayes framework. They detail the discrete Walk-Jump sampling algorithm (dWJS), which facilitates rapid, non-autoregressive sampling and can handle variable-length discrete outputs, built upon a distinct architecture for discrete EBMs. This approach facilitates the training of score-based models for discrete data, requires just a single noise level and eliminates the need for a noise schedule. As a result, the pitfalls of diffusion models, such as brittleness, training instabilities, and sluggish sampling, are sidestepped. Furthermore, the introduced method simplifies the training of EBMs by bypassing several commonly used EBM training techniques, ensuring both swift sampling and high-quality samples. The authors evaluate their method for ab initio protein discovery and design and compares against several diffusion and LLM-based models.

**Strengths:**

- The current approach builds upon foundational principles of EBMs and NEB but introduces unique methodologies for handling discrete data, formulating decoupled modeling, and tackling challenges in antibody sequence generation. In particular the study of NEB (Neural Empirical Bayes) for discrete data seems to be unique.

- The current paper distinguishes its approach from discrete diffusion models like those by Austin et al. (2023), who learn an iterative denoising process over different noise levels. Although generative modeling has been previously applied to antibodies (cited works include Shuai et al., Gligorijević et al., Ferruz & Höcker, and Tagasovska et al.), the present work highlights the unique challenges faced due to limited training data and the complexities of antibody sequences. The paper suggests that typical natural-language-based methods might struggle in this domain, pointing towards a differentiation from those methods. The current work introduces a novel formulation of decoupled energy- and score-based modeling to address the challenges in training and sampling discrete sequences, which seems distinct from previously mentioned methods.

- A new distributional conformity score (DCS) is proposed, which is useful for evaluating generated samples in comparison to a reference set. DCS provides a measure of joint distribution alignment, enabling it to capture relationships among properties rather than just aligning individual properties. This could mean that DCS provides a more comprehensive assessment of how well a generative model captures the nuances and interconnectedness of the properties within the data.

**Weaknesses:**

- While the paper touts the simplification to a single hyperparameter choice (noise level, σ) as a strength, it might also be seen as a limitation since the entire model's performance could be sensitive to this single parameter.

- Set of properties includes both continuous, binary, and discrete values and estimating the distribution by a kernel density approach may not be very effective. Also DCS may be overly influenced by outliers and extreme data points in the dataset.

**Questions:**

- Is the same value of σ=0.5 used for all reported experiments? It would be nice to see what effect lowering or increasing this value will have on some of the reported metrics for a better assessment of the sensitivity of the approach to this parameter.
- How is d obtained in this sentence? "... for the flattened sparse one-hot matrices with vocabulary size 21, d = 6237 ..." Not sure why d is so high if d=L.
- Please define the FID score (Fréchet inception distance (FID)) as well as the BLEU and add references.


I thank the authors for addressing my comments and running additional experiments to demonstrate the effect of sigma. The rebuttal addresses my concern about the DCS score. I increased my score one notch.

---

> ### Author Response · Authors · 2023-11-20
>
> We appreciate the reviewer pointing out the uniqueness of our approach and the advantages over traditional energy- and score-based approaches. The reviewer’s two main concerns relate to the effect of the noise level, sigma, and the effect of outliers on the distributional conformity score, which we address below. We hope that the reviewer will consider raising their score following our response.
>
> > While the paper touts the simplification to a single hyperparameter choice (noise level, σ) as a strength, it might also be seen as a limitation since the entire model's performance could be sensitive to this single parameter.
>
> 1. We agree that the simplification to a single hyperparameter does indeed make the model performance sensitive to this hyperparameter - we appreciate the reviewer’s point that having multiple hyperparameters gives more control over model performance. We would also like to highlight that the decoupling of the “walk” and “jump” steps hopefully alleviates this concern; while the same model can be used for both walking and jumping, it is also possible to decouple them and train two independent models with different noise levels, different architectures, and different hyperparameters. In this way, it is possible to re-introduce complexity into dWJS if desired. It would also be a very interesting future direction to adapt our approach to use a pre-trained encoder, which would introduce many additional hyperparameters. Nevertheless, we believe that there is considerable value in having a simple, robust generative model with high sample quality, and we do not see the single hyperparameter as a fundamental limitation.
>
>
> > Set of properties includes both continuous, binary, and discrete values and estimating the distribution by a kernel density approach may not be very effective. Also DCS may be overly influenced by outliers and extreme data points in the dataset.
>
> 2. The DC score is completely general and can be combined with any density estimator, so this is not a fundamental limitation of our approach. We used kernel density estimation here for convenience, and because we have few examples and low dimensions. Density models that combine continuous and discrete values that work in the low sample regime is a promising research direction. It is possible for DCS to be influenced by outliers in a dataset (and we have noted this in Appendix F), but practically speaking we do not encounter this difficulty for protein datasets because there is usually significant sequence similarity between sequences, and “outliers” generally represent out-of-distribution samples that are not “valid” protein sequences, which is exactly what we are trying to detect with the DC score. We have clarified these points in the discussion of DC score by moving some of the technical details to Appendix F to respect the page limits and adding the following text to Section 3.2: “No multiple sequence alignment or pre-processing of the sequences is required. For convenience and because we have small numbers of examples and low dimensions, we use kernel density estimation (KDE) to compute the joint density. However, DCS is completely general and can be combined with any density estimator.”
>
> > Is the same value of σ=0.5 used for all reported experiments? It would be nice to see what effect lowering or increasing this value will have on some of the reported metrics for a better assessment of the sensitivity of the approach to this parameter.
>
> 3. The sigma value is held constant, but following the reviewer’s suggestion, we have included additional experiments in Appendix A.5 and Table 6, showing the effects of changing the noise level on sample quality. Briefly, under-smoothing by setting sigma too low produces poor quality samples, while over-smoothing by setting sigma too large decreases the diversity of the samples. These intuitive results show the dependence on sigma, while also highlighting that the tradeoff between sample quality, uniqueness, and diversity, is easily controlled with a single hyperparameter, sigma. We have partially copied Table 6 with the new experimental results here for convenience:
>
> | sigma              | W_property | Unique   | E_dist   | IntDiv   |
> |--------------------|------------|----------|----------|----------|
> | 0.1                | 0.378      | 1.0 | 120.6   | 60.0     |
> | 3.0                | 0.130      | 0.995    | 44.2     | 30.0     |
>
> > How is d obtained in this sentence? "... for the flattened sparse one-hot matrices with vocabulary size 21, d = 6237 ..." Not sure why d is so high if d=L.
>
> 4. We have clarified this in the draft: the dimension for flattened one-hot matrices, d = L * v = 297 * 21 = 6237, where L is the length of the sequence and v is the vocabulary size.
>
> > Please define the FID score (Fréchet inception distance (FID)) as well as the BLEU and add references.
>
> 5. We have defined the Fréchet inception distance and the BLEU score and added appropriate references.

---

### Official Review · Reviewer_aiU9 · 2023-11-04

**Soundness:** 4 excellent
**Presentation:** 3 good
**Contribution:** 4 excellent
**Rating:** 8
**Confidence:** 4

**Summary:**

This paper introduces a new method called discrete Walk-Jump Sampling (dWJS) for generative modeling and sampling of discrete protein sequences. The key ideas are as follows. The discrete data distribution is smoothed by adding Gaussian noise, which makes it easier to model and sample from. A discrete energy-based model (dEBM) is used to learn the distribution of noisy protein sequences and is trained via contrastive divergence. A denoising model implemented as a ByteNet is separately trained to map the noisy sequences back to the original discrete space. Sampling is performed by first using Langevin MCMC to sample noisy sequences from the dEBM. Then the denoising model is used to map these noisy samples back to valid discrete protein sequences. The walk (MCMC sampling) and jump (denoising) steps are decoupled, which provides flexibility. The authors argue this provides benefits over autoregressive models, diffusion models, and traditional EBM training. The proposed method is shown to be effective on antibody protein sequence modeling and design tasks using both in silico and in vitro experiments,

Overall, I believe this is a novel contribution which demonstrates promising results on an important and challenging problem. The proposed dWJS approach is simple yet effective, providing a robust alternative to existing generative models of proteins. With additional analysis and experimental validation, this could become a leading technique for antibody design and beyond.

**Strengths:**

(+) The proposed method is intuitive and technically sound. Decoupling the sampling and denoising steps is an elegant idea.

(+)Thorough in silico evaluation using antibody-specific metrics, uniqueness, diversity, etc. In particular, a distributional conformity score  was introduced to evaluate the quality of generated protein sequences compared to a reference distribution. Extensive comparison to strong baselines.

(+) Impressive wet lab validation demonstrating high expression yields and binding rates.

**Weaknesses:**

(-) Only a single task (antibody design) is evaluated. Testing on other protein classes or discrete domains would be useful.

(-) The distributional conformity score for evaluation is introduced late with little detail. More motivation and analysis would improve clarity.
Certain details are unclear, like how sequences are aligned and handled.

**Questions:**

Your proposed approach operates purely at the sequence level, focusing specifically on antibody sequences. What are your thoughts on the relative pros and cons of sequence-only approaches compared to structure-aware approaches that leverage protein 3D structure information (e.g. inverse folding methods)? Could structure information, either from experiments or structure prediction, be incorporated to potentially improve the performance of your model? For example, might recent powerful inverse folding techniques, where they can be viewed as structure-conditional sequence generative models, like ProteinMPNN [1], ESM-IF [2], PiFold [3], or LM-Design [4] be combined with dWJS to create even more advanced antibody design frameworks? I'm curious to hear your perspective on the value of adding structural awareness and how feasible it would be to integrate with your dWJS approach.

---
[1] Dauparas, et al. Robust deep learning–based protein sequence design using ProteinMPNN. Science 2022.

[2] Hsu, et al. Learning inverse folding from millions of predicted structures. In ICML 2022

[3] Gao, et al. PiFold: Toward effective and efficient protein inverse folding. ICLR 2023.

[4] Zheng, et al. Structure-informed Language Models Are Protein Designers. ICML 2023

---

> ### Author Response · Authors · 2023-11-20
>
> We thank the reviewer for highlighting the intuitiveness and elegance of our approach, and the strength of our in silico and wet lab validation. We have addressed the reviewers’ questions in a point-by-point response below:
>
> > Only a single task (antibody design) is evaluated. Testing on other protein classes or discrete domains would be useful.
> 1. We agree with the reviewer, testing our approach on other protein classes or other discrete data settings would be interesting. We have focused on antibody design in order to present thorough and strong in silico and in vitro empirical evidence of our method’s capabilities. We have not made any assumptions unique to antibodies in developing our method, and it is a priority for future work to extend the empirical evaluation to other domains.
>
> > The distributional conformity score for evaluation is introduced late with little detail. More motivation and analysis would improve clarity. Certain details are unclear, like how sequences are aligned and handled.
>
> 2. We appreciate the reviewer’s comments related to the distributional conformity score - we have updated the text (moving some of the technical details to Appendix F to respect the page limits) to clarify the motivation for the development of DCS, sequence preparation, and intuition behind DCS. Text added to Section 3.2: “No multiple sequence alignment or pre-processing of the sequences is required. For convenience and because we have small numbers of examples and low dimensions, we use kernel density estimation (KDE) to compute the joint density. However, DCS is completely general and can be combined with any density estimator.”
>
> > Your proposed approach operates purely at the sequence level, focusing specifically on antibody sequences. What are your thoughts on the relative pros and cons of sequence-only approaches compared to structure-aware approaches that leverage protein 3D structure information (e.g. inverse folding methods)? Could structure information, either from experiments or structure prediction, be incorporated to potentially improve the performance of your model? For example, might recent powerful inverse folding techniques, where they can be viewed as structure-conditional sequence generative models, like ProteinMPNN [1], ESM-IF [2], PiFold [3], or LM-Design [4] be combined with dWJS to create even more advanced antibody design frameworks? I'm curious to hear your perspective on the value of adding structural awareness and how feasible it would be to integrate with your dWJS approach.
>
> 3. Structural data is certainly a rich source of information about proteins, and can be included into the dWJS framework in a number of ways; structure is implicitly incorporated into dWJS because antibodies are aligned according to the AHo numbering scheme. This alignment defines separate “framework” and “complementarity-determining regions” in the sequence, which have precise structural interpretations. We expect that multiple sequence alignment of most protein families already provides implicit structural information. Inverse folding models can be used to score sequence designs from dWJS if there is a known antibody structure of interest. Structure-awareness could also be more directly incorporated into our approach by adapting WJS for sampling from inverse folding models. These are promising directions for future research. Structure-aware approaches introduce additional engineering complexity and a source of experimental noise in the resolution of the crystal structure training data, which may actually degrade sequence sample quality. We also have orders of magnitude more sequence than structure data, and this is particularly acute for antibodies. For sequence design tasks, we find that sequence-only models are quite useful, but for antigen (protein target)-conditioned design where there are no known binding molecules, structure-awareness is a possible approach.

---

### Author Response · Authors · 2023-11-20

Dear Reviewers,

We thank the reviewers for their enthusiastic and supportive reviews of our paper, and the comments and questions that have led us to improve the clarity and description of technical details of our work. All reviewers agree that our work is a novel and effective generative method with strong experimental results. We have carefully considered the reviewers’ feedback and prepared a point-by-point response to each reviewer. Briefly, we have addressed major themes in the reviews related to the relationship between sequence- and structure-aware design methods, the distributional conformity score, and comparisons to baselines.

We have updated the PDF of the manuscript, reflecting the reviewer feedback, and hope the reviewers will consider increasing their scores.

---

### Meta-Review · Area_Chair_rZG7 · 2023-12-06

**Metareview:**

The paper presents a novel framework for generative modeling fo antibody sequences that combines the strengths of energy-based and score-based models to significantly improve training and sampling. The AC and reviewers all agree that this is a very exciting paper that has the potential of being a game changer the field!   The authors have done an excellent job at incorporating the reviewers points.

As future work, the AC very much encourages the authors to follow up on their points to incorporate structure awareness.

**Justification For Why Not Higher Score:**

N/A

**Justification For Why Not Lower Score:**

This is very high quality paper. It's rare to come across such a novel and pertinent approach.

---

### Decision · Program_Chairs · 2024-01-16

Accept (oral)